# Strained Conformations of Nucleosides in Active Sites of Nucleoside Phosphorylases

**DOI:** 10.3390/biom10040552

**Published:** 2020-04-05

**Authors:** Irina A. Il’icheva, Konstantin M. Polyakov, Sergey N. Mikhailov

**Affiliations:** Engelhardt Institute of Molecular Biology, Russian Academy of Sciences, Vavilov Str. 32, 119991 Moscow, Russia; imb_irina@rambler.ru (I.A.I.); kmpolyakov@gmail.com (K.M.P.)

**Keywords:** nucleoside phosphorylases, X-ray structures of nucleoside phosphorylases, nucleoside conformations

## Abstract

Nucleoside phosphorylases catalyze the reversible phosphorolysis of nucleosides to heterocyclic bases, giving α-d-ribose-1-phosphate or α-d-2-deoxyribose-1-phosphate. These enzymes are involved in salvage pathways of nucleoside biosynthesis. The level of these enzymes is often elevated in tumors, which can be used as a marker for cancer diagnosis. This review presents the analysis of conformations of nucleosides and their analogues in complexes with nucleoside phosphorylases of the first (NP-1) family, which includes hexameric and trimeric purine nucleoside phosphorylases (EC 2.4.2.1), hexameric and trimeric 5′-deoxy-5′-methylthioadenosine phosphorylases (EC 2.4.2.28), and uridine phosphorylases (EC 2.4.2.3). Nucleosides adopt similar conformations in complexes, with these conformations being significantly different from those of free nucleosides. In complexes, pentofuranose rings of all nucleosides are at the W region of the pseudorotation cycle that corresponds to the energy barrier to the N↔S interconversion. In most of the complexes, the orientation of the bases with respect to the ribose is in the high-syn region in the immediate vicinity of the barrier to *syn ↔ anti* transitions. Such conformations of nucleosides in complexes are unfavorable when compared to free nucleosides and they are stabilized by interactions with the enzyme. The sulfate (or phosphate) ion in the active site of the complexes influences the conformation of the furanose ring. The binding of nucleosides in strained conformations is a characteristic feature of the enzyme–substrate complex formation for this enzyme group.

## 1. Introduction

### 1.1. Nucleoside Phosphorylases as Key Enzymes in Nucleoside Metabolism

Nucleoside phosphorylases (NPs) catalyze the phosphorolysis of the N-glycosidic bond in purine or pyrimidine β-d-ribo-(2’-deoxyribo)nucleosides and are found in almost all organisms. This class of enzymes includes purine nucleoside phosphorylases (PNPs; EC 2.4.2.1), 5′-deoxy-5′-methylthioadenosine phosphorylases (MTAPs; EC 2.4.2.28), uridine phosphorylases (UPs; EC 2.4.2.3), pyrimidine nucleoside phosphorylases (PyNPs; EC 2.4.2.2), and thymidine phosphorylases (TPs; EC 2.4.2.4). Scheme 1 depicts the general scheme of phosphorolysis of nucleosides by NPs. The equilibrium of the reaction is shifted towards nucleosides, the shift being much more significant for PNPs as compared with UPs. Uridine phosphorylases catalyze phosphorolysis of both thymidine and uridine [1]. Purine nucleoside phosphorylases cleave the N-glycosidic bond in purine ribonucleosides and 2’-deoxynucleosides.

Reactions that are catalyzed by nucleoside phosphorylases play an important role in maintaining nucleoside homeostasis in the body, the disruption of which leads to various diseases. The essential role of these enzymes in metathesis reactions in the organisms has stimulated interest in their exploration. Numerous reviews on nucleoside phosphorylases are available. The structural organization of these enzymes [1,2], the substrate specificity and methods for the synthesis of valuable nucleosides [3,4,5,6], and metabolism in biological systems [7,8,9,10] have been considered.

The biosynthesis of nitrogenous bases and nucleosides in most of organisms can proceed by two pathways. De novo synthesis utilizes some amino acids and small organic molecules as precursors. This biochemical pathway uses numerous enzymes and it has a high requirement for energy. The steps of the de novo synthesis are virtually the same in all organisms. An alternative salvage pathway, which repeatedly utilizes fragments of RNA, is energy saving and it varies in different organisms. In all organisms, reactions catalyzed by nucleoside phosphorylases play the key role in the salvage pathway. The regulation of nucleotide metabolism and biosynthesis are considered in relation to *Escherichia coli* and *Salmonella enterica* in [9] and in [10] for mammalian.

Purinergic signaling modulates fundamental pathophysiological processes, such as tissue homeostasis, wound healing, neurodegeneration, immunity, inflammation, and cancer [11]. Many cancer processes in humans are accompanied by the disturbance of pyrimidine metabolism. Colon carcinoma cells [12], melanoma tumors [13], breast adenocarcinoma cells [14], ascites hepatoma, and Ehrlich ascites carcinoma cells [15] have increased levels of uridine phosphorylase activity. Recent experiments in mice [16] showed that the disruption of uridine homeostasis causes cancer diseases. Nucleoside phosphorylase inhibitors provide the potential to correct metabolic defects in humans [8].

These inhibitors can also be applied in the therapy of human parasitic diseases. This area of application is based on the fact that most of parasites are not able to synthesize purines and pyrimidines de novo because their genomes lack genes encoding necessary enzymes. Hence, significant differences between parasite and human nucleoside phosphorylases provide the basis for the selective inhibition of parasite enzymes. 5’-Methylthioimmucillin-H was developed as a specific inhibitor for *Plasmodium falciparum* PNP and it was proposed for the treatment of malaria [17,18]. The helminth parasite *Schistosoma mansoni* also cannot synthesize purines de novo [19]. Hence, the search for inhibitors that are specific for *Schistosoma mansoni* PNP is a challenging problem. Another important problem is the search for inhibitors specific for *Giardia lamblia* UP, because this helminth parasite is unable to synthesize pyrimidines de novo [20].

X-Ray crystallography is one of the most commonly used techniques for studying the structure-function relation of enzymes. In 1990, the first three-dimensional structure of human erythrocytic PNP (*h*PNP) was determined at 3.2 Å resolution in complex with the substrate analogue 5′-iodoformycin B [21]. The enzyme molecule was found to be composed of three subunits that were related by a crystallographic threefold axis. The substrate analogue is bound in the active site at the interface between two subunits of the trimer. The substrate binding involves seven residues of one subunit and the Phe residue of the adjacent subunit. In 1995, the X-ray structure of *E. coli* UP was determined at 2.5 Å resolution [22]. The *E. coli* UP structure is a hexamer comprised three dimers that were related by a threefold axis. Each dimer is composed of two subunits related by a non-crystallographic twofold axis. The active sites of *E. coli* UP are formed by residues belonging to two adjacent subunits of the dimer. The structure of hexameric *E. coli*. PNP was established at 2.0 Å resolution in 1997 [23]. The active site of *E. coli*. PNP also involves residues of two adjacent subunits of the dimer.

All nucleoside phosphorylases were classified into two families based on the sequence homology analysis of NPs and a comparison of their three-dimensional structures—NP-I and NP-II [1]. The NP-I family comprises hexameric and trimeric PNPs (including the MTAP subfamily) and hexameric UPs. Hexameric PNPs are enzymes from lower organisms, such as bacteria and some eukaryotic protozoa, in particular, intracellular parasites. Generally, these enzymes have a very broad spectrum of substrate specificity and are active towards both 6-amino and 6-keto derivatives of purine nucleosides [1]. Hexameric MTAPs were only isolated from the thermostable archaea *Sulfolobus solfataricus* [24], *Pyrococcus furiosus*, and *Aeropyrum pernix*. 5-Methylthioadenosine, adenosine, its 2-chloro and 2-amino derivatives, and also 6-keto derivatives of purine nucleosides (guanine and inosine) are substrates for the former two hexameric MTAPs [1]. 5′-Deoxy-5′-methylthioadenosine phosphorylase that is isolated from *Aeropyrum pernix* has some unique characteristics. On one hand, it is active towards some pyrimidines, including cytidine and deoxycytidine, and, on the other hand, unlike most of the reported MTAPs, it phosphorolyzes 2′-fluoro-modified arabinoside [25]. Trimeric PNPs and MTAPs are found in higher organisms and bacteria. Trimeric PNPs are specific for 6-keto derivatives of purine nucleosides (guanosine and inosine, but not for xanthosine), and MTA is the main substrate for trimeric MTAPs [1]. Uridine phosphorylases from bacteria and higher organisms are generally hexamers [1]. However, *Schistosoma mansoni* UP is an exception [26]. This enzyme acts as a dimer. Uridine, deoxyuridine, and thymidine are substrates for UPs [1,6,27]. Meanwhile, cytidine can also serves as a substrate for *S. cerevisiae* UP [28].

The NP-II family includes PyNPs and TPs [1]. Thymidine phosphorylase has the highest specificity among the NPs. It is only active towards thymidine and 2’-deoxyuridine. The enzymes of the NP-II family have a dimeric quaternary structure. Monomers of nucleoside phosphorylases of the NP-II family are composed of two domains, which undergo significant rearrangements during the enzymatic reaction. High-resolution X-ray diffraction data are not available for complexes of nucleosides with nucleoside phosphorylases of the NP-II family.

This review presents an analysis of conformations of nucleosides and their analogues in complexes with nucleoside phosphorylases. Only the structures determined at 2 Å resolution or higher are considered, because atomic coordinates that are derived from lower-resolution X-ray data are not reasonably accurate to establish nucleoside conformations with certainty. The available X-ray diffraction data enable the conformational analysis of nucleosides in complexes with nucleoside phosphorylases of the NP-I family.

### 1.2. Structures of the Active Sites of Nucleoside Phosphorylases of the NP-I Family

The structures of monomers of all nucleoside phosphorylases that belong to the NP-I family can be superimposed based on the spatially equivalent secondary structure elements. Figure 1 shows the superposition of the three-dimensional structures of the monomers of hexameric PNP from *B. cereus* (in complex with adenosine and phosphate, PDB 3UAW, 1.20 Å [29]), trimeric PNP from *Bos Taurus* (in complex with immucilin H and phosphate, PDB 1B80, 1.5 Å [30]), and hexameric UP from *Shewanella oneidensis* (in complex with uridine and sulfate, PDB 4R2W, 1.6 Å, [31]).

The active sites of these three enzymes are well superimposed, as can be seen in Figure 1. Figure 2, Figure 3 and Figure 4 show the active sites of these enzymes (the orientation is the same as in Figure 1).

In all enzymes, the base of the nucleoside is bound in the hydrophobic pocket and it forms hydrogen bonds with the residues of the enzyme. In all cases, adjacent subunits of the oligomeric molecules are involved in the complex formation with nucleosides. In the figures, the adjacent subunits of the hexameric PNP and UP are colored in grey. For the trimeric PNP, the adjacent subunit is only involved in the formation of the hydrophobic pocket and it is not shown in Figure 4. The ribose moiety and phosphate (or sulfate) are bound in a similar fashion in the hexameric PNP and UP. This is due to the similar structures of the phosphate- or sulfate-binding subsites of these enzymes and the presence of a conserved glutamate residue that forms hydrogen bonds with the O2′ and O3′ atoms of the ribose moiety. The radically different binding of the ribose moiety and phosphate (or sulfate) is observed in the trimeric PNP.

### 1.3. Description of Nucleoside Conformations

The parameters ***P***, ***θ_m_***, ***χ***, **ν_2_**, and **^i^***T***_j_** that are given in Tables 1–3 adequately and fully characterize nucleoside conformations. The conformation of the pentofuranose ring can be described by the following two parameters: the phase angle of pseudorotation ***P*** and the amplitude of pseudorotation ***θ_m_***. These parameters are calculated from the endocyclic torsion angles **ν_i_** by the equations given in [32]. The tables also include the endocyclic torsion angle **ν_2_** (C1′–C2′–C3′–C4′), which provides information on the degree of staggering of substituents at the C2′ and C3′ atoms of the pentofuranose ring. The mutual orientation of the pentofuranose ring and the heterocyclic base is determined by the glycosidic angle ***χ*** (C4**_pur_** (C2**_pyr_**)–N9**_pur_** (N1**_pyr_**)–C1′–O4′), according to the IUPAC–IUB rules [33]. Two regions of the angle ***χ*** are distinguished: *syn* (0° ± 90°) and *anti* (180° ± 90°). Additionally, highlight areas high *syn* (***χ*** from 90° until ~120° in *anti* region) and high *anti* (***χ*** from −90° until ~−60° in *syn* region). Scheme 2 shows these four regions of ***χ*** for uridine.

The phase angle of pseudorotation can be related to the description of the conformation of the furanose ring that is based on displacements of atoms from the mean plane of the ring. The pentofuranose ring is generally non-planar and it can adopt an envelope conformation (**^i^***E*), in which four atoms lie in a single plane, or a twist conformation (**^i^**T**_j_**), in which two adjacent atoms are displaced in the opposite directions from the mean plane of the ring. The indices correspond to the numbers of the atoms that are displaced from the mean plane of the ring, with the index 0 referring to the O4’ atom. The indices of the atoms that are displaced towards the C4’–C5’ bond are given as superscripts. The twist conformations are denoted by placing the index of the atom most displaced from the mean plane of the ring before the letter T, while the adjacent atom less displaced from the mean plane follows T.

The angle of pseudorotation of each unsymmetrical twist conformation varies over a range of 18° (Figure 5). Therefore, the character of atomic displacements in each of 20 sectors of the pseudorotation cycle is preserved, i.e., the numbers of the atoms displaced from the mean plane of the ring, the directions of these displacements, and the ratio of the displacements (in magnitude) remain unchanged. Either symmetrical twist conformations or envelope-type conformations bound the sectors of the pseudorotation cycle.

### 1.4. Conformations of Free Nucleosides

The conformations of nucleosides in the free form were studied in detail. The frequency of the occurrence of the parameters ***P*** and ***θ_m_*** in crystals of 178 nucleosides was analyzed in [34]. The phase angles ***P*** of nucleosides were found in two ranges, from −1° to 34° and from 137° to 194°. The former range belongs to the N region (***P*** varies from −45° to 45°), and the latter one to the S region (***P*** is in the range of 135–225°). Only five nucleoside derivatives were found in the range from 84° to 106°, belonging to the E region (***P*** is in the range of 45–135°). No conformers were found in the W region (***P*** varies from 225° to −45°). The amplitude of pseudorotation ***θ_m_*** of nucleosides in the crystal structures have a unimodal distribution over the range from 30° to 46°. It is almost independent of the type of the base and the phase angle ***P***. The more planar the furanose ring, the smaller the parameter ***θ_m_***.

The results of which were analyzed in [35], the N and S conformers of nucleosides exist in a dynamic equilibrium, according to extensive NMR experiments in solution. The percentage ratio of the N and S conformers existing in equilibrium in solution depends on the nature of the nucleic acid base (purine or pyrimidine) and the type of the furanose ring (ribose or deoxyribose). Purine nucleosides prefer the S conformer, while pyrimidine nucleosides prefer the N conformation. The N↔S conformational interconversions occur through the E region of the pseudorotation cycle (at about ***P*** = 90°), because the energy barrier in this region is lower. The E barrier for the N↔S interconversion occurs due to steric strain that was caused by the eclipsed orientation of substituents at the C2′ and C3′ atoms at ***P*** = 90°. For the alternative pathway through the W region in the vicinity of the phase angle ***P*** = 270°, the energy barrier is much higher. In this region, in addition to the steric strain between the substituents at the C2′ and C3′ atoms, there is also steric strain between the nitrogenous base and the exocyclic 5′-oxymethyl group due to their close arrangement of these bulky substituents.

Different computational methods evaluated the energies of barriers to pseudorotation of derivatives of the furanose ring in the W and E regions. The analysis of the results of these calculations provided the estimate of their average values [36]. The barrier of pseudorotation in the W region is ~24 kJ/mol for 2′-deoxyribofuranose and ~31 kJ/mol for ribofuranose, according to these estimations. The E region is characterized by the barrier of ~7.5 kJ/mol for 2′-deoxyribofuranose and ~16 kJ/mol for ribofuranose. The barrier of pseudorotation of purine nucleosides in the E region, which was experimentally evaluated from the ^13^C NMR data, is 20 ± 2 kJ/mol [37]. It was suggested that the energy barrier for pyrimidine nucleosides is higher than 25 kJ/mol.

In the solutions, purine nucleosides occur with equal frequency in s*yn*- and *anti*-conformations, whereas pyrimidine nucleosides more often exist in the *anti*-conformation [35,36]. The rotation about the glycosidic bond is not free [38]. In the case of pyrimidine nucleosides, the interconversions between these conformations are associated with steric strain between the protons at the C2’ and C6 atoms in the ***χ*** angle range near the high-*anti* region (from −60° to −90°) and steric interaction between the proton at the C2’ atom and O2 atom in the ***χ*** angle range in the vicinity of the high-*syn* region (90–120°). The energy of the barrier to *syn* ↔ *anti* transitions near the high-*syn* region was experimentally evaluated at 43–46 kJ/mol from the analysis of NMR spectroscopic data for pyrimidine nucleosides [39].

## 2. Discussion

### 2.1. Conformations of Nucleosides and Their Analogues in the Active Sites of PNPs and MTAP with Hexameric Quaternary Structure

Table 1 presents the conformational parameters of nucleosides in 47 complexes, including complexes with hexameric PNPs from *B. cereus*, *Vibrio cholera*, *Helicobacter pylori*, *E. coli*, *Toxoplasma gondii*, and *B. subtillis*, and complexes with hexameric MTAP from *Sulfolobus solfataricus*. In 37 complexes, the active sites contain a sulfate (phosphate) ion, and in 10 complexes this ion is absent. It should be noted that there are unacceptably short interatomic contacts and non-standard bond angles of nucleosides in the structure of *Helicobacter pylori* PNP in complex with formycin A (PDB 6F4X). Hence, we re-refined this structure to calculate the conformational parameters of the nucleoside.

Table 1 shows that the phase angles P of all the nucleosides vary over a wide part of the left part of the pseudorotation cycle (206–318°). The following two ranges of ***P*** values can be distinguished: (206–254°) and (258–318°). In the structures of the complexes containing a sulfate or phosphate ion, the nucleosides adopt a conformation that corresponds to the former range. The exceptions are two complexes of MTA with MTAP that are characterized by slightly larger angles ***P*** (259° and 267°). The conformations in the former region belong to one of the following four types: _3_T^4^, ^4^T_3_, ^4^T_O_, and _O_T^4^. Only the _3_T^4^ conformation is observed in the crystals of free nucleosides [34]. In the complexes, in which a sulfate (phosphate) ion is absent in the active site, namely, six complexes in the structure 1VHW, three complexes in the structure 6F4X (monomers 1, 2, and 4), and one of the complexes in 6F4X, in which phosphate is present with one-half occupancy (monomer 6), the angle ***P*** is in the range of 258–318°. These are the _O_T^4^, _O_T^1^, ^1^T_O_, and ^1^T_2_ conformers. These conformers were not found for nucleosides in the free form. In three of them (_O_T^4^, _O_T^1^, and ^1^T_O_), the endocyclic oxygen atom is displaced from the mean plane of the ring in the direction opposite to the glycosidic bond and the C4′–C5′ bond. This displacement of the O4′ atom (that is most pronounced in the _O_T^4^ and _O_T^1^ conformers) results in these bonds having axial positions. As mentioned above, the potential energy of these nucleoside conformations is high due to steric strain at ***P*** = 270° (W region).

In 43 complexes, purine bases adopt a high-*syn* (***χ*** is in the range of 93–120°) orientation or they are in the immediate vicinity to this conformational region (***χ*** is in the range of 127–131°). In all of these structures, two pairs of atoms of the nucleosides (N3 and C2’; C8 and O4’) form rather short contacts (about 3 Å). In the other four complexes (these are four of the seven complexes with formycin A in the structure 6F4X; Table 1, complexes 1, 2, 4, and 6), nucleosides adopt a high-*anti* orientation (***χ*** varies in the range from −67° to −71°). In these conformers, there are short contacts between the N8 and C2’ atoms (the interatomic distance is about 3 Å). No correlation between the angle ***χ*** and presence of a sulfate or phosphate ion in the complex was revealed.

Pseudorotation angles of purine nucleosides and their analogs in the active sites of hexameric PNPs are shown in Figure 5 in red in the presence of sulfate (phosphate) ions, and in green when sulfate (phosphate) ions are absent.

The puckering amplitudes ***θ_m_*** of all conformers in the active sites of hexameric PNPs and MTAP (except for the structure 4D9H) are in the range of 17–35°. In the structure 4D9H, the nucleoside adopts a conformation with a nearly planar furanose ring (***θ_m_*** = 4°), which is not typical of nucleosides. This might be attributed to the fact that the nucleoside with one-half occupancy is only present in one of the two crystallographically independent molecules. It is worth noting that not all atoms of the nucleoside are well fitted to the 2F_0_–Fc electron density at the 1σ level. Hence, the conclusions of the study [47] should be considered with caution.

### 2.2. Conformations of Nucleosides in the Active Sites of PNPs and MTAP with Trimeric Quaternary Structure

Table 2 presents the conformational parameters of nucleosides in complexes with trimeric *Mycobacterium tuberculosis*, *Schistosoma mansoni*, and calf spleen PNPs, and in the complex with trimeric MTAP from human placenta. There are a total of nine complexes, eight of which contain a sulfate (phosphate) ion.

The phase angles ***P*** of all the nucleosides in the complexes containing a sulfate (phosphate) ion are in the range of 214–243°, which is narrower than the range of ***P*** for nucleosides in complexes with hexameric PNPs, as it is seen from the Table; the angle ***χ*** is in the range of 126–133° and corresponds to the *anti* region in the immediate vicinity of the high-*syn* region. These structures also have rather short interatomic contacts similar to those found in hexameric PNPs. The puckering amplitudes ***θ_m_*** are about 25°.

In one of the three complexes of inosine with *Shistosoma mansony* PNP (PDB 3FAZ), a sulftate ion is absent. In this complex, inosine adopts another conformation: ***χ*** is in the high-*syn* region (***χ*** = 113°), the furanose ring is much more planar, and the phase angle belongs to the right part of the pseudorotation cycle (***P*** = 136°).

Pseudorotation angles of purine nucleosides and their analogs in the active sites of trimeric PNPs in Figure 5 are shown in violet in the presence of sulfate (phosphate) ions and, when sulfate (phosphate) ions are absent, they are shown in blue.

### 2.3. Conformations of Nucleosides in the Active Sites of UPs

Table 3 presents the conformational parameters of nucleosides in 14 complexes with UPs from *Vibrio cholera*, *Shewanella oneidensis*, *Schistosoma mansoni*, and *Salmonella typhimurium*. Six of these 14 complexes do not contain sulfate ions.

The data in Table 3 show that, in all six complexes of thymidine with *Vibrio cholerae* UP (PDB 4LZW) [52], sulfate ions are absent, thymidine has an *anti* orientation (***χ*** is in the range of −180°), and the furanose rings adopt _3_T^4^ or _3_T^2^ conformations (***P*** is 191–203°).

In the dimeric structure 4TXJ [26], all four crystallographically independent complexes of *Shistosoma mansoni* UP with thymidine contain the nucleoside in the immediate vicinity of the high-*syn* region (*χ* is in the range of 122–130°), and the furanose rings adopt the _O_T^4^ and _O_T^1^ conformations (*P* is in the range of 266–275°). Sulfate ions are present in all active sites.

Uridine adopts a nearly identical conformation in the active site of *Shewanella oneidensis* UP (PDB 4R2W) [31] in the presence of sulfate (*χ* = 133°, *P* = 266°, _O_T^4^). Therefore, thymidine and uridine adopt similar conformations in the presence of sulfate, i.e., the presence or the absence of the 2′-OH group in the nucleoside is not a crucial factor that is responsible for the conformation.

In Figure 5, the pseudoratation angles of uridine or thymidine in the active sites of UP in the presence of sulfate ions are shown as yellow and, when sulfate ions are absent, they are shown as light pink.

The conformation of 2,2’-O-anhydrouridine is high-*syn* (*χ =* 111°) with *P* ~230°. This conformation is observed in all three active sites in complexes with *Salmonella typhimurium* UP (PDB 3FWP) [53] and it is determined by the C2’–O–C2 covalent bond. The rigid structure of 2,2’-O-anhydrouridine is well accommodated by the active site of uridine phosphorylase. The constant Ki of this substrate analogue to UP is only eight times lower as compared to Km of Urd and 13 times lower than Km of dUrd [54].

## 3. Conclusions

The conformations of nucleosides and their analogues in complexes with nucleoside phosphorylases determined at 2 Å resolution or higher were analyzed. In all complexes with nucleoside phosphorylases, nucleosides adopt similar conformations. They are significantly different from those of free nucleosides. The pentofuranose rings of all nucleosides in complexes are located within the W region of the pseudorotation cycle. Some of them are near the energy barrier of the N↔S interconversion. The presence of sulfate (or phosphate) ions in the structures limits the range over which the conformations of the pentofuranose rings can be varied. These limitations are associated with the fact that sulfate (or phosphate) ions form additional hydrogen bonds with atoms of nucleosides in the complexes. Most of the nucleosides (both purine and pyrimidine) are in a high *syn* orientation (in the immediate vicinity of the barrier to *syn *↔ *anti* transitions at about **χ** = 120°). The *syn* orientation is energetically highly unfavorable for pyrimidine nucleosides in the free form.

To sum up, nucleosides adopt sterically strained conformations in the complexes. An increase in the potential energy of nucleosides in complexes is comparable with the energy of barriers to the N↔S interconversion through the W region and the height of the barrier to the *syn* ↔ *anti* transition in free nucleosides. The binding of nucleosides in strained conformations is a characteristic feature of the enzyme–substrate complex formation for this group of enzymes and it plays an important role in the mechanism of enzymatic catalysis.

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
