# Peer review of "Strained Conformations of Nucleosides in Active Sites of Nucleoside Phosphorylases"

_biomolecules, 2020, doi:10.3390/biom10040552_

Round 1
Reviewer 1 Report
Comments for the authors:
This is an interesting review article contributed by Dr. Mikhailov and coworkers. In this article, authors have very extensively explained different conformations of nucleosides and its analogs in the active site of nucleoside phosphorylase enzymes. Nucleoside phosphorylases catalyze the phosphorolysis of nucleosides to heterocyclic nucleoside bases and α-D-ribose 1-phosphate or α-D-2-deoxyribose 1-phosphate. Some of the nucleoside phosphorylases are drug targets against several diseases including cancer and Malaria. This review article has been written with great care and caution, however, some minor issues demand proper attention by the authors are follows.…..
- Abstract, line 13; Typo correction: ….. often elevated in tumors, which it can be used….should be ….. often elevated in tumors, which can be used……
- Figures 1, 2, 3 and 4: Please remake the figures with wall-eye stereo. Wall-eye stereo figures are better for identifying structural differences. Author can use PyMol, CCP4mg, chimera or any other program to prepare these figures.
Author Response
Comments and Suggestions for Authors
First reviewer
Comments for the authors:
This is an interesting review article contributed by Dr. Mikhailov and coworkers. In this article, authors have very extensively explained different conformations of nucleosides and its analogs in the active site of nucleoside phosphorylase enzymes. Nucleoside phosphorylases catalyze the phosphorolysis of nucleosides to heterocyclic nucleoside bases and α-D-ribose 1-phosphate or α-D-2-deoxyribose 1-phosphate. Some of the nucleoside phosphorylases are drug targets against several diseases including cancer and Malaria. This review article has been written with great care and caution, however, some minor issues demand proper attention by the authors are follows.…..
- Abstract, line 13; Typo correction: ….. often elevated in tumors, which it can be used….should be ….. often elevated in tumors, which can be used……
Line 13 The word “it” was deleted.
- Figures 1, 2, 3 and 4: Please remake the figures with wall-eye stereo. Wall-eye stereo figures are better for identifying structural differences. Author can use PyMol, CCP4mg, chimera or any other program to prepare these figures.
Figures 1, 2, 3 and 4 were redrawn as stereo figures.
Reviewer 2 Report
This review describes the Strained Conformations of Nucleosides in Active Sites of Nucleoside Phosphorylases, by authors I. Il'icheva, K. Polyakov and S. Mikhailov. It is relatively well-written manuscript but, in my opinion it is lacking in some things:
the text in some parts is not written fluently (an native English speaker check is recommended);
for example lines 47-50 too many times “review” is repeated, the sentence should be rewritten like: “The structural organization of these enzymes [1, 2], the substrate specificity and methods for the synthesis of valuable nucleosides [3–6], and the metabolism in biological systems have been considered [7–10].” The same for lines 57-59.
Some mistakes:
line 11 and 32: D-ribose, correct “D” with small cap
line 11: insert a dash between deoxyribose and 1-phosphate
line 88 and 89: delete the full stop after E. coli
The introduction is too long and the authors describe the matter as a work rather than a review; removing the introduction, figures and tables, the part of the text is small.
At present it is not suitable for publication on biomolecules.
Author Response
Comments and Suggestions for Authors
Second reviewer
This review describes the Strained Conformations of Nucleosides in Active Sites of Nucleoside Phosphorylases, by authors I. Il'icheva, K. Polyakov and S. Mikhailov. It is relatively well-written manuscript but, in my opinion it is lacking in some things:
the text in some parts is not written fluently (an native English speaker check is recommended);
for example lines 47-50 too many times “review” is repeated, the sentence should be rewritten like: “The structural organization of these enzymes [1, 2], the substrate specificity and methods for the synthesis of valuable nucleosides [3–6], and the metabolism in biological systems have been considered [7–10].” The same for lines 57-59.
lines 47-50 are rewritten: The structural organization of these enzymes [1, 2], the substrate specificity and methods for the synthesis of valuable nucleosides [3–6], and the metabolism in biological systems [7–10] have been considered.
lines 57-59 are rewritten:The regulation of nucleotide metabolism and biosynthesis are considered in [9] in relation to Escherichia coli and Salmonella enterica,andin [10] for mammalian.
Some mistakes:
line 11 and 32: D-ribose, correct “D” with small cap
line 11 and 32:the font size used for "D" has been reduced by two units.
line 11: insert a dash between deoxyribose and 1-phosphate
line 11: a dash was inserted
line 88 and 89: delete the full stop after E. coli
line 88 and 89: point was deleted after"E. coli".
The introduction is too long and the authors describe the matter as a work rather than a review; removing the introduction, figures and tables, the part of the text is small.
For the sake of clarity, we were forced to introduce definitions for the description of nucleoside conformation in the introduction. Despite the fact that the formulas for these values are not given (as the third reviewer regrets), this expands the introduction and somewhat complicates the perception of the material. However, in our opinion, the absence of these data in the text will only make understanding of the material more difficult. To reduce the size of the introduction, Figure 5 is transferred to the main part.
Reviewer 3 Report
This is a very well written, well organised review. The review introduces some basic concepts of the field before moving on to the specific focus area, namely, consideration of nucleoside structures in the context of phosphorylase enzymes, both with and without the nucleophilic phosphate(sulphate) ion. There a few minor areas that I believe should be attended to before this manuscript should be accepted for publication. These areas are summarised below:
1) The review uses extensive nomenclature/labels to describe the intricate conformational details of nucleosides. While there is clear reference to an external source for the definitions of these descriptors, it would be more convenient and make the review more complete if these definition were to be integrated into the manuscript.
2) In Scheme 1, the term Pi is defined with respect to phosphoric acid, whereas showing the monoanionic dihydrogen phosphate and diatonic mono hydrogen phosphate ions would be more informative and mechanistically correct. A similar point applies to showing the ionisation state of Rib-1-P, which should be in its mono-/dianionic form around neutral pH. It would be sensible to reference these physicochemical properties too e.g. Advances Physical Organic Chemistry 2017, 51, 187-219.
3) Scheme 2 refers to chi, however, it would be useful to show the meaning of chi on this scheme, or, more conveniently, include its alongside the other definitions discussed under point (1)
4) In Figure 4 it is difficult to differentiate yellow from gold.
5) In Figure 3, it is not very clear which part of Gly23 is being shown. I guess it is the carboxamide C (attached to red) and the alpha C....perhaps show a little more (eg alpha N) to clarify.
6) in Table 1, SO4 and PO4 are used, however, I would favour using Pi (defined in Scheme 1) and also Is, which should be defined. As it stands, these pseudo-formulae look rather odd. The 4 should be subscript in both cases, and a diatonic charge should be added to Si. As for Pi, we face the challenge of whether mono- or diatonic form should be shown, and Pi represents a simple way to get around this in light of the suggested additions to Scheme 1.
7) in line 331 the use of Ki (which also needs to be correctly type-set) is questionable. Ki is an inhibition constant, not a binding constant KD or Michaelis content Km. While there may be value in discussing and comparing these values, the fact that they are different must be considered. Furthermore, in light of the fact that the phosphorylases operate via ternary complexes (ie enzyme+Si/Pi+nucleoside), extreme care must be taken to make sure 'like is compared with like'. For example, Ki values may have been determined with one or both substrates at sub-saturation levels.
8) Minor language/editing things:
i) line 13: ... ', which can be used'...
ii) delete 'the' from line 46
iii) lines 47-50: the word reviews is used four times in quick succession, so the text becomes rather repetitive. Perhaps re-write as: Numerous reviews on nucleoside phosphorylases are available covering their structural organization [1, 2], substrate specificities [3–6], their use for the synthesis of valuable nucleosides, and their metabolism in biological systems. [7–10]....also there should be some references specifically associated with the 'synthesis of valuable nucleosides' comment.
iv) delete second 'the' from line 51
v) lines 57-59 contain 'the review' repetitively. Perhaps rewrite as: Reviews focusing on mammalian nucleoside metabolism and biosynthesis [REF] and the corresponding processes in microorganisms [REF] are available.
vi) line 78: replace 'functioning' with 'structures'
vii) lines 172-173 need to be edited for language. Perhaps: 'Additionally, for the purposes of this review, we must also highlight high syn (xxx) and high anti, because of their prevalence in phosphorylase-bound nucleoside systems.
viii) lines 188-189: '...varies over a range of 18 deg.
ix) lines 211-212—I suggest the following 'The amplitudes of pseudorotation θm of nucleosides in the crystal structures have a unimodal distribution over the range of 30o to 46o.
x) line 223-224—I suggest the following 'In this region, in addition to the steric strain between the substituents at the C2’ and C3’ atoms, there is also steric strain between the nitrogenous base and the exocyclic 5’-oxymethylene group due to their close arrangement of these bulky substituents.
xi) line 238: suggest using C6–H rather than just C6, because it will be the hydrogen that cause the steric clash.
xii) as discussed above, change SO4 and PO4 entries throughout this table.
xiii) line 257—I suggest the following: 'Note: there are two variant structures of complexes of the first monomer of structure 6F4X, which are separated by “/”.'
xiv) line 259—I suggest the following: 'Table 1 shows that the phase angles P of all nucleosides vary over a wide part of the left part of the pseudorotation cycle (206o–318o)'
xv) line 277: should C8 not more correctly be C8-H? Similarly C2'-H vs C2'
xv) lines 280-282—I think N8 should actually be C8-H and also C2'-H is the cause of the steric clash, thus, I suggest the following: 'In these conformers, there are short contacts between the C8-H and С2'-H atoms (the interatomic distance is about 3 Å). No correlation between the angle χ and the presence of a sulfate or phosphate ion in the complex was revealed.'
xvi) line 315—I suggest the following: 'The data in Table 3 show that in all six complexes of thymidine with Vibrio cholerae UP (PDB 4LZW) [54] where sulfate ions are absent, thymidine has an anti orientation (χ is in the range of –180o), and the furanose rings adopt 3T4 or 3T2 conformations (P is 191o–203o).
xvii) line 317—I suggest the following: 'These confirmations are represented by the yellow lines in Figure 5.'
xviii) lines 322, 325 and 340 change to plural i.e. 'conformations'
xix) line 326 '...group in the nucleoside....'
xx) line 330 change 'well accommodates' to 'is well accommodated by'
xxi) line 337 change 'at' to 'within'
xxii) line 339 change 'range in which' to 'range over which'
xxiii) line 352 to '...providing us with the program for calculating the parameters for pseudorbtation within the pentofuranose cycle'
xxiv) The references need some editing eg journal titles need to be abbreviated, capitalised etc. Several references all need to be complete and full eg, complete refs 6, title of 34, authors in 40
Author Response
Comments and Suggestions for Authors
Third reviewer
This is a very well written, well organised review. The review introduces some basic concepts of the field before moving on to the specific focus area, namely, consideration of nucleoside structures in the context of phosphorylase enzymes, both with and without the nucleophilic phosphate(sulphate) ion. There a few minor areas that I believe should be attended to before this manuscript should be accepted for publication. These areas are summarised below:
1)The review uses extensive nomenclature/labels to describe the intricate conformational details of nucleosides. While there is clear reference to an external source for the definitions of these descriptors, it would be more convenient and make the review more complete if these definition were to be integrated into the manuscript.
In our opinion, it is not necessary to give formulas for determining the conformational parameters of nucleosides, since this will unjustifiably increase the volume of introduction (it should be noted that the second reviewer proposes a reduction in introduction).
2) In Scheme 1, the term Pi is defined with respect to phosphoric acid, whereas showing the monoanionicdihydrogen phosphate and diatonic mono hydrogen phosphate ions would be more informative and mechanistically correct. A similar point applies to showing the ionisation state of Rib-1-P, which should be in its mono-/dianionic form around neutral pH. It would be sensible to reference these physicochemical properties too e.g. Advances Physical Organic Chemistry 2017, 51, 187-219.
The legend for Scheme 1 was modified: “Scheme 1. Phosphorolysis of nucleosides catalyzed by NPs. Urd – uridine, UP- uridinephosphorylase, PNP – purine nucleoside phosphorylase, Rib-1-P – α-D-ribose-1-phosphate, Pi – inorganic phosphate ion.”
3) Scheme 2 refers to chi, however, it would be useful to show the meaning of chi on this scheme, or, more conveniently, include its alongside the other definitions discussed under point (1)
The definition of the value of chi is given on the lines 170-171
4) In Figure 4 it is difficult to differentiate yellow from gold.
Yellow is selected in Figures 1-4 to indicate sulfur atoms. In our opinion, the difference in yellow and gold colors in Figure 2 is quite noticeable.
5) In Figure 3, it is not very clear which part of Gly23 is being shown. I guess it is the carboxamide C (attached to red) and the alpha C....perhaps show a little more (eg alpha N) to clarify.
Figures 2-4 are redrawn in stereo. This allows you to clearly trace the interatomic bonds in active centers.
6) in Table 1, SO4 and PO4 are used, however, I would favour using Pi (defined in Scheme 1) and also Is, which should be defined. As it stands, these pseudo-formulae look rather odd. The 4 should be subscript in both cases, and a diatonic charge should be added to Si. As for Pi, we face the challenge of whether mono- or diatonic form should be shown, and Pi represents a simple way to get around this in light of the suggested additions to Scheme 1.
According to 2):The legend for Scheme 1 was modified: Pi – is the notation of inorganic phosphate ion.
7) in line 331 the use of Ki (which also needs to be correctly type-set) is questionable. Ki is an inhibition constant, not a binding constant KD or Michaelis content Km. While there may be value in discussing and comparing these values, the fact that they are different must be considered. Furthermore, in light of the fact that the phosphorylases operate via ternary complexes (ieenzyme+Si/Pi+nucleoside), extreme care must be taken to make sure 'like is compared with like'. For example, Ki values may have been determined with one or both substrates at sub-saturation levels.
In the cited paper [56] the following procedures were used for the estimation of KMandKI.
Michaelis constant (KM) was estimated for phosphorolysis of uridine and its derivatives. The constant KM of phosphorolysis of uridine by UP was determined from the spectrophotometric estimation of the reaction rate at 280 nm. The substrate concentration varied in the range of 10−200 μM for uridine derivatives at a constant phosphate concentration (50 mM), pH 7.5, at 25°C. The reactions were initiated by adding 0.03 units of the enzyme; the rate was measured for 5 min. To obtain KM,weused nonlinear regression analysis of the Michaelis–Menten and Lineweaver–Burk plots.
KI of uridine derivatives were measured spectrophotometrically.Anenzyme was preincubated at various concentrations of an inhibitor at 25°Cfor15 min, and the initial rate was measured for phosphorolysis of the natural substrate.Substrateswereusedat0.5 or 1 KM. The estimation of KI followed the Dixon method.
The text in line 331 was corrected. Now it is :“The constant Ki of this substrate analogue to UP is only eight times lower compared to Km of Urd and 13 times lower than Km of dUrd [56].”
8) Minor language/editing things:
- i) line 13: ... ', which can be used'...
Word “It” was deleted.
- ii) delete 'the' from line 46
Word “The” in line 47(previously it was line 46) was deleted.
iii) lines 47-50: the word reviews is used four times in quick succession, so the text becomes rather repetitive. Perhaps re-write as: Numerous reviews on nucleoside phosphorylases are available covering their structural organization [1, 2], substrate specificities [3–6], their use for the synthesis of valuable nucleosides, and their metabolism in biological systems. [7–10]....also there should be some references specifically associated with the 'synthesis of valuable nucleosides' comment.
lines 47-50 are rewritten: The structural organization of these enzymes [1, 2], the substrate specificity and methods for the synthesis of valuable nucleosides [3–6], and the metabolism in biological systems [7–10] have been considered.
- iv) delete second 'the' from line 51
Second 'the' from line 51 was deleted.
- v) lines 57-59 contain 'the review' repetitively. Perhaps rewrite as: Reviews focusing on mammalian nucleoside metabolism and biosynthesis [REF] and the corresponding processes in microorganisms [REF] are available.
lines 57-59 are rewritten:The regulation of nucleotide metabolism are considered in [9] in relation to Escherichia coli and Salmonella enterica,andin [10] for mammalian.
- vi) line 78: replace 'functioning' with 'structures'
line 78: was corrected as: word “functioning” was replaced to "structure - function relation"
vii) lines 172-173 need to be edited for language. Perhaps: 'Additionally, for the purposes of this review, we must also highlight high syn (xxx) and high anti, because of their prevalence in phosphorylase-bound nucleoside systems.
Text was not been corrected. These addition regions were defined for nucleoside. For example see [35].
viii) lines 188-189: '...varies over a range of 18 deg.
In line 188 the words “in the” were replaced to “over a”.
- ix) lines 211-212—I suggest the following 'The amplitudes of pseudorotationθm of nucleosides in the crystal structures have a unimodal distribution over the range of 30o to 46o.
Line 211-212 the word “has” was replaced to “structures have” and the word “and is in” was replaced to “over”. (Now it is Line 204.)
- x) line 223-224—I suggest the following 'In this region, in addition to the steric strain between the substituents at the C2’ and C3’ atoms, there is also steric strain between the nitrogenous base and the exocyclic 5’-oxymethylene group due to their close arrangement of these bulky substituents.
Line 223 The sentence “In this region, apart from the steric strain between the substituents at the C2’ and C3’ atoms, there is steric strain between the nitrogenous base and the exocyclic 5’-oxymethyl group due to the close arrangement of these bulky substituents at P= 270o.´was replaced to ”In this region, in addition to the steric strain between the substituents at the C2’ and C3’ atoms, there is also steric strain between the nitrogenous base and the exocyclic 5’-oxymethyl group due to their close arrangement of these bulky substituents.” (Now it is Line 216 – 219).
- xi) line 238: suggest using C6–H rather than just C6, because it will be the hydrogen that cause the steric clash.
Text was not been corrected. Because the sentence contained the word “protons”.
xii) as discussed above, change SO4 and PO4 entries throughout this table.
Text was not been corrected. See parts 2 and 6.
xiii) line 257—I suggest the following: 'Note: there are two variant structures of complexes of the first monomer of structure 6F4X, which are separated by “/”.'
The sentence line 257 ”Note: there are two variants structure of complexes in the first monomer of the structure 6F4X. They are shown through “/”.” was replaced to “Note: there are two variant structures of complexes of the first monomer of structure 6F4X, which are separated by “/”.”
xiv) line 259—I suggest the following: 'Table 1 shows that the phase angles P of all nucleosides vary over a wide part of the left part of the pseudorotation cycle (206o–318o)'
The sentence line 259 “As it is seen from the Table 1 the phase angles P of all nucleosides vary in a wide range in the left part of the pseudorotation cycle (206o–318o).” was changed to “Table 1 shows that the phase angles P of all nucleosides vary over a wide part of the left part of the pseudorotation cycle (206o–318o)”
- xv) line 277: should C8 not more correctly be C8-H? Similarly C2'-H vs C2'
The proposal was rejected because hydrogen atoms are not localized in the protein structures and all discussions about short contacts are carried out for heavy atoms.
- xv) lines 280-282—I think N8 should actually be C8-H and also C2'-H is the cause of the steric clash, thus, I suggest the following: 'In these conformers, there are short contacts between the C8-H and С2'-H atoms (the interatomic distance is about 3 Å). No correlation between the angle χ and the presence of a sulfate or phosphate ion in the complex was revealed.'
The proposal was rejected because hydrogen atoms are not localized in the protein structures and all discussions about short contacts are carried out for heavy atoms. By the way, in these complexes is formycin A, which contain in 8-position N.
xvi) line 315—I suggest the following: 'The data in Table 3 show that in all six complexes of thymidine with Vibrio cholerae UP (PDB 4LZW) [54] where sulfate ions are absent, thymidine has an anti orientation (χ is in the range of –180o), and the furanose rings adopt 3T4 or 3T2 conformations (P is 191o–203o).
Line 315 the words“ As it is seen from the Table 3” was replaced to “The data in Table 3 show that”
xvii) line 317—I suggest the following: 'These confirmations are represented by the yellow lines in Figure 5.'
The sentence line 317 ” They are shown as yellow zone at the Figure 5.” was replaced to “These conformations are represented by the light pink lines in Figure 5.”
xviii) lines 322, 325 and 340 change to plural i.e. 'conformations'
Line 322, 325, 340 the word “conformation” was replaced to “conformations”
xix) line 326 '...group in the nucleoside....'
Line 326 the words “group in nucleoside” was replaced to “group in the nucleoside”
- xx) line 330 change 'well accommodates' to 'is well accommodated by'
Accepted
xxi) line 337 change 'at' to 'within'
Accepted
xxii) line 339 change 'range in which' to 'range over which'
Accepted
xxiii) line 352 to '...providing us with the program for calculating the parameters for pseudorbtation within the pentofuranose cycle'
Accepted
xxiv) The references need some editing eg journal titles need to be abbreviated, capitalised etc. Several references all need to be complete and full eg, complete refs 6, title of 34, authors in 40
All references are edited.
Round 2
Reviewer 2 Report
nothing to report